# Mechanism of superconductivity and electron-hole doping asymmetry in $\kappa$-type molecular conductors

Hiroshi Watanabe [1,2], Hitoshi Seo[1,3] & Seiji Yunoki[1,3,4]

Unconventional superconductivity in molecular conductors is observed at the border of metal-insulator transitions in correlated electrons under the influence of geometrical frustration. The symmetry as well as the mechanism of the superconductivity (SC) is highly controversial. To address this issue, we theoretically explore the electronic properties of carrier-doped molecular Mott system $\kappa$-(BEDT-TTF)$_2$X. We find significant electron-hole doping asymmetry in the phase diagram where antiferromagnetic (AF) spin order, different patterns of charge order, and SC compete with each other. Hole-doping stabilizes AF phase and promotes SC with $d_{xy}$-wave symmetry, which has similarities with high-$T_c$ cuprates. In contrast, in the electron-doped side, geometrical frustration destabilizes the AF phase and the enhanced charge correlation induces another SC with extended-$s + d_{x^2-y^2}$ wave symmetry. Our results disclose the mechanism of each phase appearing in filling-control molecular Mott systems, and elucidate how physics of different strongly-correlated electrons are connected, namely, molecular conductors and high-$T_c$ cuprates.

[1] RIKEN Cluster for Pioneering Research (CPR), Wako, Saitama 351-0198, Japan. [2] Waseda Institute for Advanced Study, Waseda University, Shinjuku, Tokyo 169-8050, Japan. [3] RIKEN Center for Emergent Matter Science (CEMS), Wako, Saitama 351-0198, Japan. [4] RIKEN Center for Computational Science (R-CCS), Kobe, Hyogo 650-0047, Japan. Correspondence and requests for materials should be addressed to H.W. (email: h-watanabe@riken.jp)

Understanding the intimate correlation among metal-insulator (MI) transition, magnetism, and superconductivity (SC) is one of the most challenging issues in modern condensed matter physics. The most well-studied example is the high-$T_c$ cuprates, where SC is observed when mobile carriers are doped into the parent antiferromagnetic (AF) Mott insulators[1–3]. A general understanding there, supported by various experiments and theories, is that strong AF spin fluctuation mediates the $d$-wave SC that appears through the filling-control Mott MI transition generating mobile charge carriers. However, can we export this mechanism to other strongly correlated materials? To address this question, it is crucial to make a comparison among different classes of materials. In this respect, heavy fermion compounds and molecular conductors provide such opportunities[4–8].

The family of quasi two-dimensional molecular conductors $\kappa$-(ET)$_2X$ (ET = BEDT-TTF, and $X$ takes different monovalent anions[9]) is in fact compared often with the cuprates[10]. They indeed have common factors: simple quasi-two-dimensional electronic structure to begin with in the non-interacting limit and the Mott MI transition and SC closely related with each other. However, there are important differences: First, in $\kappa$-(ET)$_2X$, SC appears through the bandwidth-control Mott transition; the carrier density is usually unchanged but the pressure (either physically or chemically) is the controlling factor. Although the variation of the carrier density is necessary for direct comparisons, it has not been realized in $\kappa$-(ET)$_2X$ for a long time due to experimental difficulties. Second, while the cuprates are basically governed by the physics nearby 1/2-filling, $\kappa$-(ET)$_2X$ is a 3/4-filled system. The similarity enters when the so-called dimer approximation is applied in the latter[11], resulting in the effective 1/2-filled system (dimer model). Although the dimer model has been extensively studied using various theoretical methods[12–25], the validity of the dimer approximation itself is recently reexamined[26–32]. Especially, the importance of the intradimer charge degree of freedom[33,34] and intersite Coulomb interactions[27,29], which are discarded in the dimer approximation, has attracted much attention because of recent experimental suggestions[35–38]. Third, in $\kappa$-(ET)$_2X$, SC is observed not only next to the AF insulators but also to nonmagnetic (candidate of gapless spin-liquid) insulators[8,39–41]. The strong influence of geometrical frustration owing to the anisotropic triangular arrangement of dimers is present in this family.

Recently, carrier doping has been realized either chemically in $\kappa$-(ET)$_4$Hg$_{3−\delta}Y_8$ ($Y$ = Br or Cl)[42–45] or in $\kappa$-(ET)$_2$Cu[N(CN)$_2$]Cl ($\kappa$-Cl) by using electric-double-layer transistor (EDLT) technique[46,47], revealing intriguing phenomena such as a dome-shaped SC region, anomalous metallic behaviors, and significant electron-hole doping asymmetry, which are all reminiscent of the high-$T_c$ cuprates. Therefore, $\kappa$-type ET systems can now provide a unique playground of both filling-control and bandwidth-control Mott transitions with SC phases nearby, for which a unified theoretical understanding is highly desired.

In this paper, we theoretically study the ground-state properties of $\kappa$-(ET)$_2X$ varying the carrier number from 3/4-filling, in order to elucidate the electronic phases appearing near the Mott transition in this system, especially SC, and to investigate their stabilities beyond mean field treatments. The intradimer charge degree of freedom and intersite Coulomb interactions are explicitly considered. We find that the ground-state phase diagram shows significant electron-hole asymmetry in the stability of AF phase and in terms of competing two types of SC. While in the hole-doped side $d_{xy}$-wave SC is favored by the AF spin fluctuation as in the high-$T_c$ cuprates, the electron doping highlights the geometrical frustration and the charge degree of freedom, which are unique in $\kappa$-(ET)$_2X$, stabilizing extended-$s$ + $d_{x^2−y^2}$-wave SC.

The electron-hole doping asymmetry, including the symmetry of SC, is attributed to the degree of frustration that is controlled by carrier doping. This is a conceptually new perspective to $\kappa$-(ET)$_2X$, which can also be applied to other frustrated systems in general. Our results, beyond the usual description based on the 1/2-filled dimer model, thus provide new understanding of how physics of molecular conductors and high-$T_c$ cuprates are distinct.

## Results

**Model derivation and framework.** The electronic properties of molecular conductors are modeled by a simple model where the molecules are replaced by lattice sites[48]. They are described by the extended Hubbard model (EHM)[11,49], a textbook model for studying correlated electrons. The Hamiltonian is given as

$$H = -\sum_{\langle i,j \rangle \sigma} t_{ij}(c^\dagger_{i\sigma}c_{j\sigma} + \text{H.c.}) + U\sum_i n_{i\uparrow}n_{i\downarrow} + \sum_{\langle i,j \rangle} V_{ij}n_i n_j, \quad (1)$$

where $c^\dagger_{i\sigma}$ ($c_{i\sigma}$) is a creation (annihilation) operator of electron at molecular site $i$ with spin $\sigma(=\uparrow,\downarrow)$, $n_{i\sigma} = c^\dagger_{i\sigma}c_{i\sigma}$, and $n_i = n_{i\uparrow} + n_{i\downarrow}$. $U$ and $V_{ij}$ are on-site and intersite Coulomb repulsions, respectively. $<i, j>$ denotes pairs of neighboring molecules in the $\kappa$-type geometry, labeled by $b_1$, $b_2$, $p$, and $q$, as shown in Fig. 1a.

The tight-binding parameters $t_{ij}$ are set for the deuterated $\kappa$-(ET)$_2$Cu[N(CN)$_2$]Br ($\kappa$-Br), which locates very close to the MI transition[50], and are adopted from a first-principles band calculation as $(t_{b_1}, t_{b_2}, t_p, t_q) = (196, 65, 105, -39)$ meV $= (1.0, 0.332, 0.536, -0.199)\, t_{b_1}$[51]. We set the largest hopping integral $t_{b_1}$ as the unit of energy. The unit cell is a rectangle with $R_x \times 2R_y$ and $\boldsymbol{\delta}_\pm = (\delta_x, \pm\delta_y)$ are vectors connecting the centers of

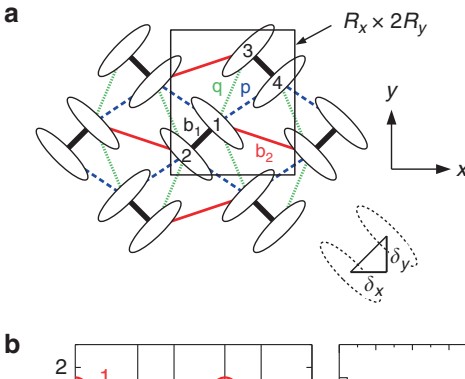

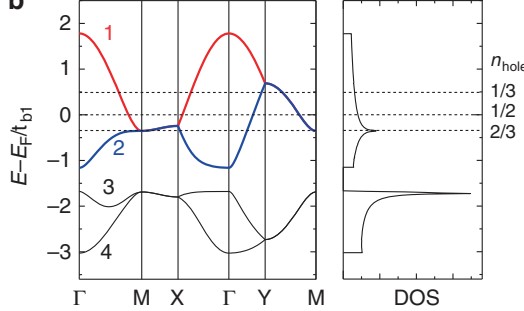

**Fig. 1** Lattice and electronic structures of the model. **a** Two-dimensional lattice structure of $\kappa$-(ET)$_2X$. Unit cell (black rectangle) contains four molecules labeled as 1–4 and its size is $R_x \times 2R_y$. Molecules are connected with bonds $b_1$, $b_2$, $p$, and $q$. The centers of the dimers (1–2 and 3–4) form an anisotropic triangular lattice. **b** Band structure and density of states (DOS) of $\kappa$-(ET)$_2$Cu[N(CN)$_2$]Br. $E_F$ denotes the Fermi energy for $n_{\text{hole}} = 1/2$ (undoped case). Dotted lines correspond to the Fermi energy for $n_{\text{hole}} = 1/3$, 1/2, and 2/3. High symmetry points of momentum **k** are Γ(0,0), M($\pi/R_x$, $\pi/2R_y$), X($\pi/R_x$,0), and Y(0, $\pi/2R_y$) (See also Fig. 3b)

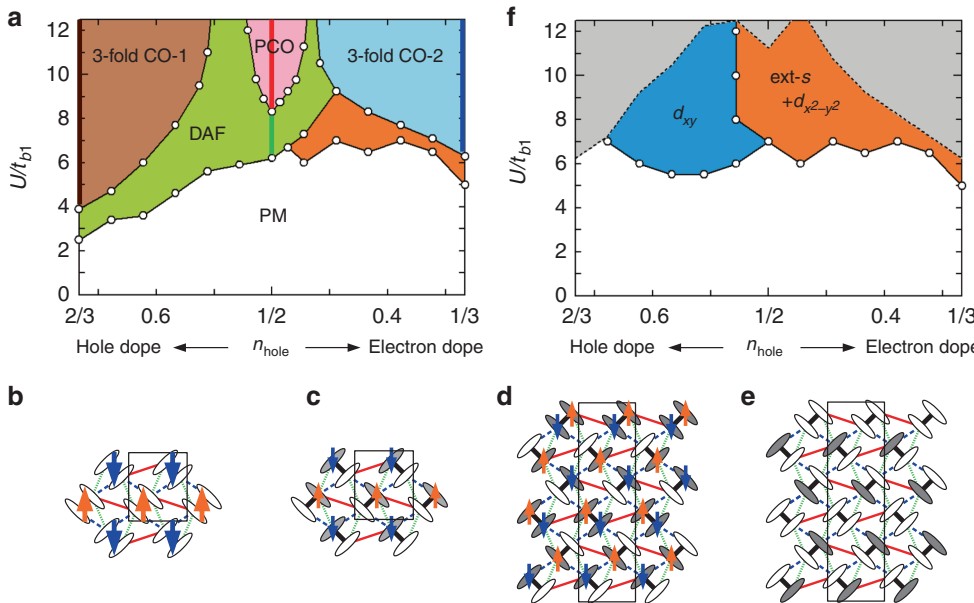

**Fig. 2 Ground-state phase diagram. a** Ground-state phase diagram of the EHM for $\kappa$-(ET)$_2$X. DAF, PCO, 3-fold CO-1, and 3-fold CO-2 phases are insulating only along vertical bold lines and metallic for other colored regions. **b**–**e** Schematic view of each symmetry-broken phase: (**b**) DAF, (**c**) PCO, (**d**) 3-fold CO-1 (magnetic), and (**e**) 3-fold CO-2 (nonmagnetic). Up (down) arrows represent up (down) spin-rich molecular sites. Solid (open) ovals represent hole-rich (hole-poor) molecular sites. The black rectangles are unit cells. **f** Ground-state phase diagram for SC, ignoring other ordered phases. The gray regions indicate where the strong intersite Coulomb interactions strongly suppress the hole mobility (see the text)

molecules facing each other in a dimer (see Fig. 1a). Here, we set $(R_x, R_y, \delta_x, \delta_y) = (1.0, 0.7, 0.3, 0.3)R_x$ with $R_x$ as a unit of length[29]. The non-interacting band structure is shown in Fig. 1b. Among the four energy bands, the upper two bands (bands 1 and 2) contribute to form the Fermi surface (FS). For the undoped case, the electron density per molecular site is $n = 3/2$ (3/4-filling) and it corresponds to the hole density $n_{hole} = 2 - n = 1/2$. In this study, we change $n_{hole}$ from 1/3 to 2/3 to investigate the doping dependence of the system. The corresponding Fermi energies are indicated in Fig. 1b by dotted lines.

The effect of Coulomb interactions is treated using a variational Monte Carlo (VMC) method[52–54]. The trial wave function considered here is a Gutzwiller–Jastrow type, $|\Psi\rangle = P_{J_c} P_{J_s} |\Phi\rangle$. $|\Phi\rangle$ is a one-body part constructed by diagonalizing the one-body Hamiltonian, and $P_{J_c}$ and $P_{J_s}$ are charge and spin Jastrow factors, respectively. The explicit form of them are described in Methods. In the following, we show results for 1152 molecular sites (corresponding to $L = 24$, see in Methods), which is large enough to avoid finite size effects.

**Ground-state phase diagram.** Figure 2a shows the ground-state phase diagram. The hole density $n_{hole} = 2 - n$ and the on-site Coulomb interaction $U/t_{b_1}$ are varied as parameters, while the ratio between $U$ and the largest intersite Coulomb interaction $V_{b_1}$ is fixed at $V_{b_1}/U = 0.50$. The other intersite Coulomb interactions are set as $(V_{b_2}, V_p, V_q) = (0.56, 0.66, 0.58) V_{b_1}$, assuming the $1/r$-dependence. At $n_{hole} = 1/2$ (undoped case)[29], a first-order phase transition occurs, with increasing $U/t_{b_1}$, from a paramagnetic metal (PM) to a dimer-type AF (DAF) phase in which the spins between dimers order in a staggered way as shown in Fig. 2b. This transition corresponds to the Mott MI transition. As $U/t_{b_1}$ increases further, there appears a polar charge-ordered (PCO) phase breaking the inversion symmetry[31,34] with AF spin order, which can avoid the energy loss of $V_{b_1}$, $V_{b_2}$, and $V_p$, at the expense of the energy loss of $V_q$ as shown schematically in Fig. 2c. The DAF and PCO phases are insulating at $n_{hole} = 1/2$. They have

also been found in previous studies for the 3/4-filled Hubbard models[11,31,49] and the effective strong coupling models[33,34], and are stabilized in the relevant parameter regions for $\kappa$-(ET)$_2$X. Experimentally, the DAF phase is widely observed in $\kappa$-(ET)$_2$X as AF dimer-Mott insulator and the PCO phase is proposed to be related to the dielectric anomaly observed in $\kappa$-(ET)$_2$Cu$_2$(CN)$_3$ ($\kappa$-CN)[36] and the insulating phase in $\kappa$-(ET)$_2$Hg(SCN)$_2$Cl[55]. Note that SC is a metastable state for $U/t_{b_1} = 7$–11.5, lying on each side of the DAF-PCO boundary.

Away from $n_{hole} = 1/2$, significant doping asymmetry is observed and several different phases appear. For the hole-doped side ($n_{hole} > 1/2$), while the PCO phase is rapidly suppressed, the DAF phase is enhanced to a smaller $U/t_{b_1}$ region toward $n_{hole} = 2/3$. Note that in these phases the system becomes metallic once the doping is finite. Furthermore, a 3-fold charge-ordered (3-fold CO-1) phase appears for larger $U/t_{b_1}$. The 3-fold CO-1 phase shows charge disproportionation and magnetic order as shown in Fig. 2d; hole-rich sites form a two-dimensional network with AF spin order. This phase is insulating at $n_{hole} = 2/3$ (along the brown line in Fig. 2a) since the electron density fits the commensurability, and metallic for other hole densities because the excess holes can move through the ordered holes.

For the electron-doped side ($n_{hole} < 1/2$), the situation is much different. The PCO and DAF (both become metallic) phases are rapidly suppressed and another CO (3-fold CO-2) phase and a SC phase appear. The pattern of the charge disproportionation is opposite to that in the 3-fold CO-1 phase (hole rich ↔ hole poor) as shown in Fig. 1e; this configuration can fully avoid the intersite Coulomb interactions. Similar to the 3-fold CO-1 phase, the 3-fold CO-2 phase is insulating at $n_{hole} = 1/3$ (along the blue line in Fig. 2a) and metallic for other hole densities. Note that the 3-fold CO-2 phase is stabilized also for $n_{hole} = 1/2$ when $V_{b_1}/U$ is larger ($\geq 0.55$)[29,31] and is smoothly connected in the parameter space. While the 3-fold CO-1 phase is accompanied by the magnetic order, 3-fold CO-2 is nonmagnetic. We have tried several magnetic ordering patterns that coexist with 3-fold CO-2. However, none of them are stabilized because the CO pattern

in the hole-rich sites forms a triangular-like structure and the spin degree of freedom is fully frustrated, and furthermore the distances between hole-rich sites are much longer than the original intermolecular bonds and therefore the effective magnetic exchange couplings are quite small. For smaller $U/t_{b_1}$, the SC phase is realized by doping, located between the DAF/3-fold CO-2 and PM phases. The symmetry of the SC is the extended-$s + d_{x^2-y^2}$-wave type, same with the one shown in our previous study at $n_{hole} = 1/2$[29]. Details are discussed later.

Although SC does not appear as the ground state in the hole-doped side, we find finite superconducting condensation energy in the phase diagram. Figure 2f shows the region where the condensation energy is finite, ignoring other ordered phases by setting Weiss fields to be zero in $|\Phi\rangle$. It is possible that the hidden SC phase appears if the DAF phase is destabilized by, e.g., disorder effect associated with doping or phase separation. Therefore, it is worthwhile to study the most favored SC phase even if it is a metastable state. While the $d_{xy}$-wave SC is dominant for most of the hole-doped side, the extended-$s + d_{x^2-y^2}$-wave SC is stabilized for the electron-doped side. Namely, the symmetry of SC changes with carrier doping. Note that the charge correlation is greatly enhanced toward regions indicated by gray shade in Fig. 2f. In these regions, the mobility of holes are strongly restricted due to the strong intersite Coulomb interactions, and stable VMC simulations are difficult unless additional Weiss fields that induce long-range CO are introduced in $|\Phi\rangle$.

**Fermi surface and spin structure factor.** The electron-hole doping asymmetry is closely related to the shape of the FS and the interdimer magnetic fluctuations. Figure 3a–c show the non-interacting FS for $n_{hole} = 2/3$, 1/2, and 1/3. As $n_{hole}$ increases from 1/2 (hole doping), the FS shifts toward the right and left edges of the first Brillouin zone (M-X line in Fig. 3b). Since the energy gap of the DAF order opens along the Brillouin zone edge[11], the DAF order becomes more favored for hole doping. For $n_{hole} = 2/3$, the FS almost touches the Brillouin zone edge, and there the DAF region extends down to $U/t_{b_1} \sim 2.5$, as shown in Fig. 2a. Note that the Fermi energy is located in the vicinity of van Hove singularity at $n_{hole} = 2/3$ as shown in Fig. 1b. Around this hole density, anomalous behavior such as pseudogap phenomena is naively expected[46]. In clear contrast, the FS departs from the M-X

line for electron doping, consistent with the tendency of the DAF order being rapidly suppressed for $n_{hole} < 1/2$.

Next, Fig. 3d–f show the interdimer spin structure factor defined as,

$$S^{dim}(\mathbf{q}) = \frac{1}{N_{dim}} \sum_{l,m} \langle M_l^{dim} M_m^{dim} \rangle e^{i\mathbf{q}\cdot(\mathbf{r}_l - \mathbf{r}_m)}, \quad (2)$$

for $n_{hole} = 2/3$, 1/2, and 1/3. Here, $N_{dim}(=L^2)$ is the total number of dimers and $M_l^{dim} = (n_{2l-1\uparrow} + n_{2l\uparrow}) - (n_{2l-1\downarrow} + n_{2l\downarrow})$ is the total spin density within $l$-th dimer formed by molecular sites $2l - 1$ and $2l$ with the central position $\mathbf{r}_l$[56]. Since the dimer centers form the anisotropic triangular lattice[11], the corresponding first Brillouin zone is the anisotropic hexagon. As shown in Fig. 3d, $S^{dim}(\mathbf{q})$ for $n_{hole} = 2/3$ peaks around $(0, \pm\pi/R_y)$, which corresponds to the DAF spin configuration, suggesting that the AF spin fluctuation is enhanced by the Coulomb interactions and thus the DAF order is favored. On the other hand, for $n_{hole} = 1/2$, the peaks appear around six vertices of the Brillouin zone, as shown in Fig. 3e. This implies that the spin structure becomes more triangular-lattice like (frustrated) and the AF spin fluctuation is suppressed as compared with that at $n_{hole} = 2/3$. For $n_{hole} = 1/3$, the peak structures almost diminish (see Fig. 3f), and the DAF order is not stabilized around this hole density.

**Superconducting gap functions.** The above mentioned electron-hole asymmetry in the spin and the charge degrees of freedoms are the keys to understand the competition between the two types of SC. The main contribution of the gap function for the $d_{xy}$-wave SC is given as

$$\Delta^\alpha = \Delta_1^\alpha \left[ \cos\left(\frac{1}{2}k_x R_x + k_y R_y\right) - \cos\left(\frac{1}{2}k_x R_x - k_y R_y\right) \right], \quad (3)$$

where $\alpha(=1, 2)$ denotes a band index, and $\Delta_i^\alpha$ is the pairing with the $i$-th neighbor dimers in the real space and treated as a variational parameter. We optimize the real space pairing up to 22nd neighbor dimers and find that the overall feature of the gap function is determined within the fourth neighbor, i.e., $\Delta_m^\alpha$ for $m \le 4$. The term contaning $\Delta_1^\alpha$ in Eq. (3) gives nodes in the horizontal (along $k_x$-axis) and vertical (along $k_y$-axis) directions. This is because the two diagonal pairings (orange and blue bars in Fig. 4a) have different sign, giving a $d_{xy}$-type contribution. Therefore, this gap symmetry is referred to as $d_{xy}$-wave[26]. Note that the terms corresponding to real space pairings parallel to horizontal (along $x$-axis) and vertical (along $y$-axis) directions vanish since they are along the nodal directions. This SC phase has been discussed in analogy with that of high-$T_c$ cuprates, ascribing the diagonal directions in $\kappa$-type structure to an approximate square lattice[10]. As in the case for high-$T_c$ cuprates, the sign of the gap function on the FS changes four times (see Fig. 4c).

On the other hand, the main contribution of gap function for the extended-$s+d_{x^2-y^2}$-wave SC is given as

$$\Delta^\alpha = \Delta_1^\alpha \left[ \cos\left(\frac{1}{2}k_x R_x + k_y R_y\right) + \cos\left(\frac{1}{2}k_x R_x - k_y R_y\right) \right] \\ + \Delta_2^\alpha \cos k_x R_x + \Delta_3^\alpha \cos 2k_y R_y, \quad (4)$$

The first term contaning $\Delta_1^\alpha$ in Eq. (4) (blue bars in Fig. 4b) does not change sign within the first Brilloin zone and becomes zero only along the zone boundary; this term gives an extended-$s$-like contribution. Furthermore, $\Delta_1^\alpha$ changes sign between different bands, namely, $sgn\Delta_1^1 = -sgn\Delta_1^2$. In this respect, the pairing symmetry can also be referred to as $s_\pm$, similar to that of iron-based SC[57,58]. The second and third terms containing $\Delta_2^\alpha$ and $\Delta_3^\alpha$ in Eq. (4) (orange and yellow bars in Fig. 4b, respectively) give nodes in the diagonal direction; these terms give a $d_{x^2-y^2}$-like

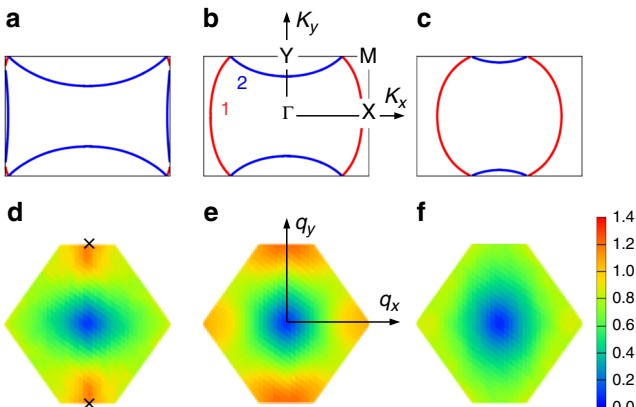

**Fig. 3** Fermi surface and spin structure factor. **a–c** Non-interacting ($U = V_{ij} = 0$) FS for (**a**) $n_{hole} = 2/3$, (**b**) 1/2, and (**c**) 1/3. Red and blue indicate the portions of the FS formed by bands 1 and 2, respectively (See Fig. 1b). High symmetry points are $\Gamma(0,0)$, M($\pi/R_x$, $\pi/2R_y$), X($\pi/R_x$,0), and Y(0, $\pi/2R_y$). **d–f** Interdimer spin structure factor $S^{dim}(q)$ for (**d**) $n_{hole} = 2/3$ with $U/t_{b_1} = 4$, (**e**) $n_{hole} = 1/2$ with $U/t_{b_1} = 7$, and (**f**) $n_{hole} = 1/3$ with $U/t_{b_1} = 6$. Crosses indicate $\mathbf{q} = (0, \pm\pi/R_y)$ in **d**

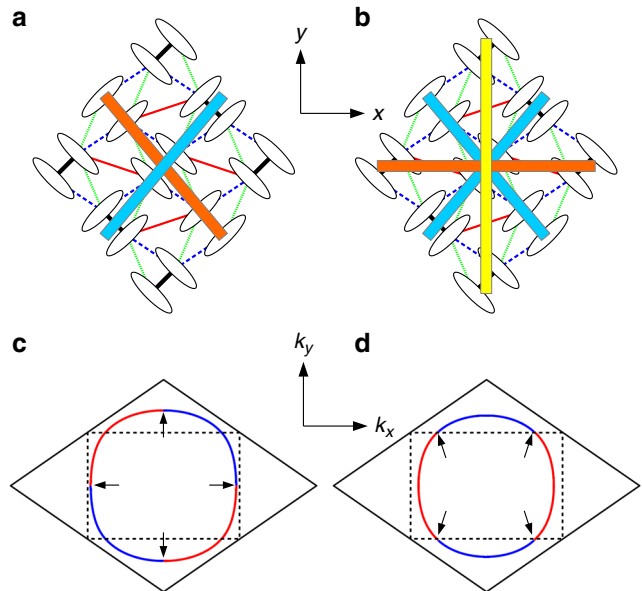

**Fig. 4** Symmetry of superconducting gap functions. Schematic real space pairing of the $d_{xy}$-wave (**a**) and the extended-$s + d_{x^2-y^2}$-wave (**b**), and sign changes of the gap function on the FS for the $d_{xy}$-wave (**c**) and the extended-$s + d_{x^2-y^2}$-wave (**d**). Red (blue) in **c** and **d** represents plus (minus) sign of the gap function and arrows indicate node points on the FS. The dotted rectangle is the original first Brillouin zone and the solid rhombus is the unfolded Brillouin zone when the dimer model is considered

contribution. The gap symmetry is thus referred to as an extended-$s + d_{x^2-y^2}$-wave[26,28,32,59]. The sign changes of the gap function on the FS is shown in Fig. 4d.

The competition between the two types of SC has been already discussed for the undoped case. First, in the effective 1/2-filled dimer model, many early studies inferred the $d_{xy}$-wave type[12–14,16–18]. However, its stability over the AF phase has recently been doubted in several numerical studies including the VMC approach[20,22]. Kuroki et al.[26] were the first to point out the importance of treating the 3/4-filled model, namely, considering the intradimer charge degree of freedom. In the 3/4-filled Hubbard model, the two types of SC compete and either of them is favored depending on the parameters, especially the degree of dimerization[26,28,32]. It is only recently that the importance of the intersite Coulomb interaction $V_{ij}$ on SC was pointed out[27,29]. In fact, our previous VMC study for the undoped case found that, although both symmetries are enhanced by the intersite Coulomb interactions, the extended-$s + d_{x^2-y^2}$-wave is slightly more favored[29].

As shown in Fig. 2f, our calculations find that this competition is released when the mobile carriers are doped into the system. For the hole-doped side, the AF spin fluctuation toward the DAF order is enhanced due to the shape of the FS and the Coulomb interactions, as already seen in Fig. 3, and consistent with the recent calculations based on the dimer model[46]. Similarly to the high-$T_c$ cuprates, the AF spin fluctuation mediated SC is then developed with the strong singlet correlation along diagonal bonds shown in Fig. 4a, resulting in the $d_{xy}$-wave symmetry. On the other hand, the AF spin fluctuation is suppressed with electron doping as seen in Fig. 3f and the spin singlet correlation along horizontal and vertical bonds compete with that along diagonal bonds. The extended-$s + d_{x^2-y^2}$-wave is eventually favored since all bonds can contribute to the singlet pairing. The carrier doping deforms the shape of the FS and modifies the AF spin fluctuation, thus inducing the change of the symmetry of

SC. Although the electron-hole asymmetry shown in Fig. 2a appears similar to that of high-$T_c$ cuprates at a glance, they are much different especially for the SC phase. In both cases, the van Hove singularity appears only in the hole-doped side, causing the electron-hole asymmetry. However, different physics are delicately involved in $\kappa$-(ET)$_2$X, as we have shown so far, and even the change of the symmetry of SC occurs by carrier doping. This is due to the unique geometrical frustration inherent in the triangular-like lattice structure of $\kappa$-(ET)$_2$X. Furthermore, this asymmetry is expected to be robust for $\kappa$-(ET)$_2$X in general because the van Hove singularity is always located in the hole-doped side for the realistic parameter set of these conductors.

We can show more directly how the spin and charge correlations are correlated to the stability of the SC phases. The interdimer charge structure factor is defined as

$$N^{\dim}(\mathbf{q}) = \frac{1}{N_{\dim}} \sum_{l,m} \langle n_l^{\dim} n_m^{\dim} \rangle e^{i\mathbf{q}\cdot(\mathbf{r}_l - \mathbf{r}_m)}, \quad (5)$$

where $n_l^{\dim} = (n_{2l-1\uparrow} + n_{2l\uparrow}) + (n_{2l-1\downarrow} + n_{2l\downarrow})$ is the total charge density within $l$-th dimer formed by molecular sites $2l - 1$ and $2l$ with the central position $\mathbf{r}_l$[56]. Figure 5a–d show $S^{\dim}(\mathbf{q})$ and $N^{\dim}(\mathbf{q})$ for $n_{\text{hole}} = 640/1152 = 0.556$ and $544/1152 = 0.472$, where the $d_{xy}$-wave and the extended-$s + d_{x^2-y^2}$-wave SC are stabilized, respectively. For $n_{\text{hole}} = 0.556$, $S^{\dim}(\mathbf{q})$ peaks around $(0, \pm\pi/R_y)$ that are favorable for the $d_{xy}$-wave SC, while for $n_{\text{hole}} = 0.472$, $S^{\dim}(\mathbf{q})$ shows frustrated spin structure that are favorable for the extended-$s + d_{x^2-y^2}$-wave SC. These are consistent with the mechanism of SC described above. On the other hand, $N^{\dim}(\mathbf{q})$ peaks around $(0, \pm 3\pi/2R_y)$ for both $n_{\text{hole}} = 0.556$ and 0.472. This is because they are located near the instability of 3-fold CO-1 and 2, respectively, and the corresponding wave vectors are the same.

The superconducting condensation energy $\Delta E$ and spin/charge correlations are contrasting between the two SC phases. Figure 5e, f show the $U/t_{b_1}$ dependence of $N_{\text{3-fold}}$, $S_{\text{peak}}$, and $\Delta E$. $N_{\text{3-fold}} = N^{\dim}(0, \pm 3\pi/2R_y)$ and $S_{\text{peak}} = S^{\dim}(0, \pm\pi/R_y)$ for $n_{\text{hole}} = 0.556$. The peak position in $S^{\dim}(\mathbf{q})$ changes along the upper and lower edges of the Brillouin zone for $n_{\text{hole}} = 0.472$ and we take the maximum value for $S_{\text{peak}}$. For $n_{\text{hole}} = 0.556$, $\Delta E$ does enhance, but despite the more rapid increase of $N_{\text{3-fold}}$ toward 3-fold CO-1 instability, it rather follows $S_{\text{peak}}$. This is consistent with the usual AF spin fluctuation picture discussed in high-$T_c$ cuprates. On the other hand, for $n_{\text{hole}} = 0.472$, $\Delta E$ is greatly enhanced following the rapid increase of $N_{\text{3-fold}}$, which suggests the close correlation between the extended-$s + d_{x^2-y^2}$-wave SC and the 3-fold CO-2 type charge fluctuation.

## Discussion

Let us note that the intersite Coulomb interactions $V_{ij}$ are indispensable to the stability of SC, not only for the extended-$s + d_{x^2-y^2}$-wave SC phase but also for the metastable $d_{xy}$-wave SC phase. As we have shown in the previous work[29], no long-range ordered phases are stabilized in the absence of $V_{ij}$ for the undoped condition (or unphysically large $U$ is necessary). This is also the case in the doped condition studied here. This indicates that in $\kappa$-(ET)$_2$X, the charge degree of freedom is still active even with large $U$ and the cooperation between $U$ and $V_{ij}$ induces various phases such as the DAF, PCO, 3-fold COs, and SC. Especially, the extended-$s + d_{x^2-y^2}$-wave SC is enhanced toward both polar and 3-fold type CO instabilities. Recent experiment on photoinduced phase transition in $\kappa$-Br suggests that the polar charge oscillation is enhanced near the superconducting transition[60], consistent with our picture that the SC is enhanced toward the CO instability.

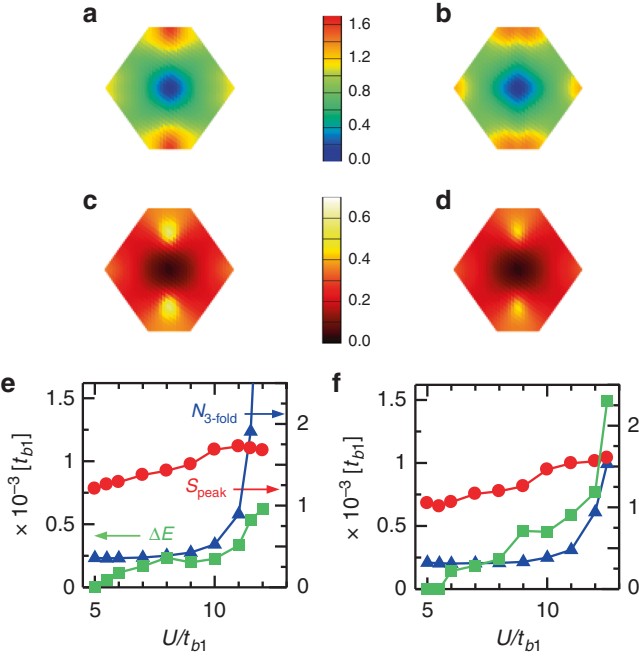

**Fig. 5 Spin and charge structure factors, and superconducting condensation energies.** $S^{\mathrm{dim}}(\mathbf{q})$ for $n_{\mathrm{hole}} = 0.556$ (**a**) and 0.472 (**b**). $N^{\mathrm{dim}}(\mathbf{q})$ for $n_{\mathrm{hole}} = 0.556$ (**c**) and 0.472 (**d**). $U/t_{b_1} = 10$ is set for all. $U/t_{b_1}$ dependence of $N_{\mathrm{3\text{-}fold}}$ (blue triangles), $S_{\mathrm{peak}}$ (red circles), and $\Delta E$ (green squares) for $n_{\mathrm{hole}} = 0.556$ (**e**) and 0.472 (**f**). The symmetry of SC is the $d_{xy}$-wave (extended-$s + d_{x^2-y^2}$-wave) for $n_{\mathrm{hole}} = 0.556$ (0.472). The statistical errors of the Monte Carlo sampling are within the size of the symbols in **e** and **f**

Experimentally, the symmetry of the SC for the undoped case is still controversial[38,61–67]. This is consistent with our result suggesting that the competition between the two types of SC is most pronounced near the undoped $n_{\mathrm{hole}} = 1/2$. Slight modification of parameters may alter the stability of the two. Now in $\kappa$-Cl, both electron and hole doping are realized using the EDLT[46,47]. The filling-control MI transition and the emergence of SC have been confirmed with significant electron-hole doping asymmetry. Our result suggests that the different SC appears by carrier doping; the $d_{xy}$-wave SC on the hole-doped side and the extended-$s + d_{x^2-y^2}$-wave SC on the electron-doped side. The former is due to the strong AF spin fluctuation, similar to high-$T_c$ cuprates, and the latter is the consequence of frustrated spin structure and enhanced charge correlation under the geometrical frustration characteristic of the $\kappa$-type ET compounds. Experiments for the chemically doped $\kappa$-(ET)$_4$Hg$_{3-\delta}$Y$_8$ suggest similarities with the high-$T_c$ cuprates, especially for the non-Fermi-liquid behaviors above the SC transition temperature[42–44]. This is overall consistent with our results because this system is hole-doped. However, the tight-binding parameters for this system are rather closer to the isotropic triangular-lattice like arrangement of dimers, where the AF phase is expected to be unfavored because of the stronger geometrical frustration. Indeed, the temperature dependence of the spin susceptibility is similar to that of $\kappa$-CN[45], which does not show any magnetic long-range orders. It is difficult to fully explain the character of $\kappa$-(ET)$_4$Hg$_{3-\delta}$Y$_8$ within the present study and the analysis with appropriate tight-binding parameters is necessary for further understanding.

The nonmagnetic (candidate of gapless spin-liquid) insulator and neighboring SC observed in $\kappa$-CN are also an intriguing phenomena for the undoped case. Although the tight-binding parameters and the resulting electronic structure of $\kappa$-CN is different from the present study, the intradimer charge degree of freedom and the intersite Coulomb interactions should play

crucial roles as pointed out previously[29]. Indeed, the anomalous dielectric response[36] and Raman spectroscopy[37] indicate that the intradimer charge degree of freedom is active in $\kappa$-CN. The stability of spin-liquid and SC phase in the 1/2-filled Hubbard model on the anisotropic triangular lattice, which is the approximate dimer model of $\kappa$-CN, is still controversial despite the long and extensive studies[12–15,17,19–25]. The analysis for the 3/4-filled EHM employed in this study will be an alternative way to investigate this issue and it is left for future studies.

## Methods

**Details of the VMC method.** Here, we show the details of the VMC method. Trial wave function is a Gutzwiller-Jastrow type, $|\Psi\rangle = P_{J_c} P_{J_s} |\Phi\rangle$. $|\Phi\rangle$ is a one-body part constructed by diagonalizing the one-body Hamiltonian including the off-diagonal elements $\{D\}$, $\{M\}$, and $\{\Delta\}$ to treat long-range orders of charge, spin, and SC, respectively. The renormalized hopping integrals $(\tilde{t}_{b_1}, \tilde{t}_{b_2}, \tilde{t}_p, \tilde{t}_q)$ are also included in $|\Phi\rangle$ as variational parameters, where $\tilde{t}_{b_1} = t_{b_1}$ is fixed as a unit. $P_{J_c} = \exp\left[-\sum_{i,j} v_{ij}^c n_i n_j\right]$ and $P_{J_s} = \exp\left[-\sum_{i,j} v_{ij}^s s_i^z s_j^z\right]$ are charge and spin Jastrow factors which control long-range charge and spin correlations, respectively. Here, $s_i^z = n_{i\uparrow} - n_{i\downarrow}$, and we assume $v_{ij}^c = v^c(|r_i - r_j|)$ and $v_{ij}^s = v^s(|r_i - r_j|)$, where $r_i$ is the position of molecular site $i$. The variational parameters in $|\Psi\rangle$ are $\tilde{t}_{b_2}$, $\tilde{t}_p$, $\tilde{t}_q$, $\{D\}$, $\{M\}$, $\{\Delta\}$, $\{v_{ij}^c\}$, and $\{v_{ij}^s\}$, and they are simultaneously optimized using the stochastic reconfiguration method[68]. The total number of molecular sites is $4 \times L \times L/2 = 2L^2$ and varied from $L = 12$ to $L = 24$ with antiperiodic boundary conditions in both $x$ and $y$ directions of the primitive lattice vectors (see Fig. 1a).

The one-body part $|\Phi\rangle$ for the PM, DAF, and PCO states can be obtained by diagonalizing the one-body Hamiltonian,

$$\tilde{H}_0 = \sum_{\mathbf{k}\sigma} \left(c_{\mathbf{k}1\sigma}^\dagger, c_{\mathbf{k}2\sigma}^\dagger, c_{\mathbf{k}3\sigma}^\dagger, c_{\mathbf{k}4\sigma}^\dagger\right)$$
$$\times \begin{pmatrix} -s_\sigma M_1^z & T_{21}^* & T_{31}^* & T_{41}^* \\ T_{21} & -s_\sigma M_2^z & T_{32}^* & T_{42}^* \\ T_{31} & T_{32} & s_\sigma M_3^z & T_{43}^* \\ T_{41} & T_{42} & T_{43} & s_\sigma M_4^z \end{pmatrix} \begin{pmatrix} c_{\mathbf{k}1\sigma} \\ c_{\mathbf{k}2\sigma} \\ c_{\mathbf{k}3\sigma} \\ c_{\mathbf{k}4\sigma} \end{pmatrix} \quad (6)$$

$$= \sum_{\mathbf{k}\alpha\sigma} \tilde{E}_\alpha(\mathbf{k}) a_{\mathbf{k}\alpha\sigma}^\dagger a_{\mathbf{k}\alpha\sigma} \quad (7)$$

with the hopping matrix elements given as

$$T_{21} = -\tilde{t}_{b_1} e^{-i(k_x \delta_x + k_y \delta_y)} - \tilde{t}_{b_2} e^{i\{k_x(R_x - \delta_x) - k_y \delta_y\}}, \quad (8)$$

$$T_{31} = -\tilde{t}_q\left[e^{i\{k_x(R_x/2 - \delta_x) + k_y R_y\}} + e^{i\{k_x(R_x/2 - \delta_x) - k_y R_y\}}\right]$$
$$= -2\tilde{t}_q e^{ik_x(R_x - \delta_x)} \cos k_y R_y, \quad (9)$$

$$T_{32} = -2\tilde{t}_p e^{-ik_y(R_y - \delta_y)} \cos\frac{1}{2}k_x R_x, \quad (10)$$

$$T_{41} = -2\tilde{t}_p e^{ik_y(R_y - \delta_y)} \cos\frac{1}{2}k_x R_x, \quad (11)$$

$$T_{42} = -2\tilde{t}_q e^{-ik_x(R_x/2 - \delta_x)} \cos k_y R_y, \quad (12)$$

$$T_{43} = -\tilde{t}_{b_1} e^{i(k_x \delta_x - k_y \delta_y)} - \tilde{t}_{b_2} e^{-i\{k_x(R_x - \delta_x) + k_y \delta_y\}}, \quad (13)$$

where $c_{\mathbf{k}m\sigma}^\dagger$ ($c_{\mathbf{k}m\sigma}$) is a creation (annihilation) operator of electron at molecule $m$ ($=1$–4), as indicated in Fig. 1a, with momentum $k$ and spin $\sigma(=\uparrow, \downarrow)$, and $s_\sigma = 1$ ($-1$) for $\sigma = \uparrow(\downarrow)$. $M_m^z$ is a variational parameter which induces the staggered AF long-range order aligned to the $z$ direction for molecule $m$; $M_1^z = M_2^z = M_3^z = M_4^z = 0$ for the PM state, $M_1^z = M_2^z = M_3^z = M_4^z \neq 0$ for the DAF state, and $M_1^z = M_3^z \neq M_2^z = M_4^z$ for the PCO state. $a_{\mathbf{k}\alpha\sigma}^\dagger$ ($a_{\mathbf{k}\alpha\sigma}$) in Eq. (7) is a creation (annihilation) operator of quasiparticle in band $\alpha$ with momentum $k$ and spin $\sigma(=\uparrow, \downarrow)$, obtained by diagonalizing Eq. (6), and $\tilde{E}_\alpha(\mathbf{k})$ is the corresponding quasiparticle energy.

For the 3-fold CO-1 and 2 states, the unit cell is three times larger than the original one and contains 12 molecules, as shown in Fig. 2d, e. Therefore, $|\Phi\rangle$ is obtained by diagonalizing a $12 \times 12$ matrix. The corresponding Weiss field is $D_{m\sigma} \sum_{\mathbf{k}\sigma} c_{\mathbf{k}m\sigma}^\dagger c_{\mathbf{k}m\sigma}$, which induces charge disproportionation and spin ordering within the unit cell through the variational parameters $D_{m\sigma}$ for $m = 1, 2, \ldots, 12$.

Finally, $|\Phi\rangle$ for SC is obtained by diagonalizing the BCS-type mean-field Hamiltonian,

$$\tilde{H}_{BCS} = \sum_{\mathbf{k}} \left( a^\dagger_{\mathbf{k}1\uparrow}, a^\dagger_{\mathbf{k}2\uparrow}, a^\dagger_{\mathbf{k}3\uparrow}, a^\dagger_{\mathbf{k}4\uparrow} a_{-\mathbf{k}1\downarrow}, a_{-\mathbf{k}2\downarrow}, a_{-\mathbf{k}3\downarrow}, a_{-\mathbf{k}4\downarrow} \right)$$

$$\times \begin{pmatrix} \xi_1 & 0 & 0 & 0 & \Delta^1 & 0 & 0 & 0 \\ 0 & \xi_2 & 0 & 0 & 0 & \Delta^2 & 0 & 0 \\ 0 & 0 & \xi_3 & 0 & 0 & 0 & \Delta^3 & 0 \\ 0 & 0 & 0 & \xi_4 & 0 & 0 & 0 & \Delta^4 \\ \Delta^1 & 0 & 0 & 0 & -\xi_1 & 0 & 0 & 0 \\ 0 & \Delta^2 & 0 & 0 & 0 & -\xi_2 & 0 & 0 \\ 0 & 0 & \Delta^3 & 0 & 0 & 0 & -\xi_3 & 0 \\ 0 & 0 & 0 & \Delta^4 & 0 & 0 & 0 & -\xi_4 \end{pmatrix} \begin{pmatrix} a_{\mathbf{k}1\uparrow} \\ a_{\mathbf{k}2\uparrow} \\ a_{\mathbf{k}3\uparrow} \\ a_{\mathbf{k}4\uparrow} \\ a^\dagger_{-\mathbf{k}1\downarrow} \\ a^\dagger_{-\mathbf{k}2\downarrow} \\ a^\dagger_{-\mathbf{k}3\downarrow} \\ a^\dagger_{-\mathbf{k}4\downarrow} \end{pmatrix}, \quad (14)$$

where $\Delta^\alpha$ denotes gap function for band $\alpha$ and $\xi_\alpha = \tilde{E}_\alpha(\mathbf{k}) - \tilde{\mu}$ is a quasiparticle energy measured from the renormalized chemical potential $\tilde{\mu}$. $\Delta^\alpha = \Delta^\alpha(\{\Delta^\alpha_i\})$ is constructed from the real space pairing $\Delta^\alpha_i$ up to the 22nd neighbor dimers ($i = 1, 2, \ldots, 22$). Since the band 3 and 4 are located much below the Fermi energy (see Fig. 1b), their contribution to the pairing is negligible and therefore we set $\Delta^3 = \Delta^4 = 0$.

## Data availability
The data that support the findings of this study are available from the corresponding author on reasonable request.

## Code availability
The code that support the findings of this study are available from the corresponding author on reasonable request.

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

## Acknowledgements

The authors thank R. Kato, Y. Kawasugi, H. Itoh, S. Iwai, Y. Kawakami, and M. Naka for useful discussions. The computation has been done using the facilities of the Super-computer Center, Institute for Solid State Physics, University of Tokyo. This work has been supported by JSPS KAKENHI (Grant Nos 26800198, 26400377, 16H02393, and 18H01183) and Waseda University Grant for Special Research Projects (Project Number: 2018B-352).

## Author contributions

H.W. performed the VMC simulations and prepared the figures. Results were analyzed and the paper was written by H.W., H.S., and S.Y.

## Additional information

**Competing interests:** The authors declare no competing interests.

