## [Peer Review File · Nature Communications]

Reviewers' comments:

Reviewer #1 (Remarks to the Author):

The authors investigated the extended Hubbard model with a lattice that mimics the kappa-type ET-based conductors by means of a variational Monte Carlo technique. The novelty of this paper is to reveal the phase diagram with doping level as a parameter and finding three kinds of charge ordered phases, a dimer AF phase and two kinds of superconducting phases, one of which was predicted to be hidden behind the AF phase. In most organic (super)conductors, carrier doping has been out of control. Recently, however, two types of doped Mott insulators have been available; naturally occurring doped compounds and artificially carrier-doped systems. So, this theoretical work is timely and, in my opinion, will offer a theoretical ground for further discussing the experimental results of the doped organic systems.

In modelling the kappa-type compounds, a key is to employ a dimer lattice with on-site and off-site Coulomb repulsions, which gives fluctuations in both of spin and charge channels. Experimentally, the importance of the spin fluctuations has been well recognized, but it is only recently that the charge fluctuations are argued to be vital in this family of materials. Therefore, the interplay between the spin and charge fluctuations are of keen interest. The present work has revealed a map of this interplay in the parameter space of correlation versus doping level in both of normal and superconducting states. The prediction of the electron-hole asymmetry in doping is quite interesting and will give insight into the existing experimental results.

The manuscript is intelligibly written. The model includes many parameters such as several transfer integrals and Coulomb interactions, U and V ; however, the authors neatly fix the parameters and relate U and V to pick up the essential features of the model. I think that this manuscript deserves publication in Nature Communications.

A minor comment--- can authors comment on the magnetic ground state of 3-fold CO-2?

Kazushi Kanoda

Reviewer #2 (Remarks to the Author):

This manuscript by Watanabe et al. employed a microscopic model to investigate the pairing symmetry of unconventional superconductivity in molecular conductors kappa-(BEDT-TTF)₂X in both electron-doped and hole-doped sides. It is currently an important issue due to the experimental realization of superconductivity in this family of strongly correlated systems. The microscopic model employed by the authors is the extended Hubbard model with both onsite and intersite Coulomb interactions. By doing variationally Monte Carlo calculations, they found apparent asymmetry between the electron-doped and hole-doped sides of the phase diagram. The pairing symmetry is d_{xy} in the hole-doped side while it is extended $s+d_{x^2-y^2}$ in the electron-doped side. They also provided some explanations about the asymmetry. To me, this work is important and timely due to the experimental relevance although the method and model employed are not novel. I intend to recommend its publication in Nat. Comm. after the authors could address the comments/questions I have below.

1) In the hole-doped side, superconductivity is not the most favored state in the model studied by the authors. I wonder if the SC with d_{xy} pairing symmetry can be realized as a ground state by using another set of parameters in the extended Hubbard model.

2) In the electron-doped side, the SC pairing symmetry is extended $s+d_{x^2-y^2}$ which breaks both charge- $U(1)$ symmetry and lattice rotational symmetry. I wonder if the authors could comment on its finite temperature behaviors, namely whether there are two transitions or one transition as the

temperature is lowered.

3) To what extent the asymmetry between hole-doping and electron-doping depends on the model parameters. Namely, is this asymmetry robust enough?

Reviewer #3 (Remarks to the Author):

summary:

Based on Variational Monte Carlo (VMC) calculations the authors present a theoretical analysis of the phase diagram of the extended Hubbard model on a two-dimensional lattice representing the lattice structure of kappa-(ET)₂X molecular conductors. The authors investigate the appearance of various ground-state phases as a function of onsite-Coulomb-repulsion strength and electron filling. Metallic, superconducting and insulating phases of various kinds (antiferromagnetic and different type of charge ordered phases) are presented and discussed. The authors point to the asymmetry in the phase diagram between hole doping and electron doping the lattice and its relation to recent experimental observations in kappa-(ET)₂X systems where carrier doping was achieved either chemically (Ref. 40-43 in the manuscript) or by field-effect doping (Ref. 44, 45 in the manuscript).

originality and discussion:

the family of ET charge transfer salts in a kappa arrangement has been experimentally and theoretically intensively studied as a representative of Mott-Hubbard physics with a rich phase diagram including Mott insulator, bad metal and superconductor phases as a function of chemical and/or external pressure. More recently the possibility of carrier doping, as mentioned above, has evidenced a significant electron-hole doping asymmetry. The extended Hubbard model, as studied by the authors, has been proposed in the past by many authors, including some of the authors of the manuscript, to be an adequate model to study these materials and there exist already many investigations on the ground-state phases of this model for various interaction strengths (onsite and nearest-neighbor) performed by some of the authors as well as by other groups by means of VMC and a few other many-body methods. The novelty of the present manuscript is that the authors extend the study to carrier filling away from the 3/4 filling that covers regions in phase space now accessible to experiment and they observe an electron-hole asymmetry.

Whether these results are enough to publish the manuscript as a nature communications is not completely clear. I view the present manuscript as an extension of previous results that would be appropriate in a more specialized journal.

The manuscript is well written, however there are a few issues to consider:

- while the authors cite previous works related to the model considered and results obtained, there is no discussion on the comparison of phases obtained in the present work with respect to what other groups have done. In other words, the cited work is not commented or compared although the authors are using similar methods to some of them and explore similar phase regions.

How do their results compare with previous ones, where are the similarities and where are the

differences in the common regions studied?

- how would temperature affect the results?

- the authors mention that superconductivity is not the ground-state in the hole-doped side. Which is the ground state? and how is then the phase diagram modified?

Response to Reviewer Comments

We would like to first thank all reviewers for their valuable and insightful comments and suggestions to our manuscript. We have properly addressed their comments and revised the manuscript accordingly. Our responses to all the reviewers' comments follow the list of changes given below.

List of Changes

- (1) We have corrected and added the sentences “The electron-hole doping ...” starting from the 2nd line from the bottom of right-hand side of page 1 to the end of the paragraph, of the new manuscript (likewise in the following).
- (2) We have added the two sentences “They have also ...” at the 3rd line from the top of the left-hand side of page 3.
- (3) We have added the sentence “Note that in ...” at the 2nd line from the bottom of the left-hand side of page 3.
- (4) We have added the sentences “While the 3-fold CO-1 ...” starting from the 3rd line from the top of the left-hand side of page 4, to the 13th line from the top of the left-hand side of page 4. “(magnetic)” and “(nonmagnetic)” are also added for the explanation of (d) and (e) in the caption of Figure 2.
- (5) We have added two sentences “It is possible that ...” at the 8th line from the bottom of the left-hand side of page 4.
- (6) We have added the sentences “First, in the effective ...” starting from the 12th line from the bottom of the right-hand side of page 5, to “... doped into the system.” at the 8th line from the top of the left-hand side of page 6.
- (7) We have added two sentences “In both cases, ...” at the 27th line from the top of the left-hand side of page 6, and a sentence “Furthermore, ...” at the end of this paragraph.
- (8) We have added references 16, 18, and 55.
- (9) We have corrected sentences “Experiments for the ...” starting from the 27th line from the bottom of the left-hand side of page 7 to the end of the paragraph for better description.
- (10) We have corrected minor typos and improved several sentences for better description

Reviewer #1 (Remarks to the Author):

The authors investigated the extended Hubbard model with a lattice that mimics the kappa-type ET-based conductors by means of a variational Monte Carlo technique. The novelty of this paper

is to reveal the phase diagram with doping level as a parameter and finding three kinds of charge ordered phases, a dimer AF phase and two kinds of superconducting phases, one of which was predicted to be hidden behind the AF phase. In most organic (super)conductors, carrier doping has been out of control. Recently, however, two types of doped Mott insulators have been available; naturally occurring doped compounds and artificially carrier-doped systems. So, this theoretical work is timely and, in my opinion, will offer a theoretical ground for further discussing the experimental results of the doped organic systems.

We appreciate the reviewer's positive comment about the novelty of this paper. We agree with the reviewer that the recent realization of carrier doping into organic conductors will open up a frontier in strongly correlated electron systems. As suggested by the reviewer, we indeed expect that our work will provide a theoretical guideline for further experimental developments.

In modelling the kappa-type compounds, a key is to employ a dimer lattice with on-site and off-site Coulomb repulsions, which gives fluctuations in both of spin and charge channels. Experimentally, the importance of the spin fluctuations has been well recognized, but it is only recently that the charge fluctuations are argued to be vital in this family of materials. Therefore, the interplay between the spin and charge fluctuations are of keen interest. The present work has revealed a map of this interplay in the parameter space of correlation versus doping level in both of normal and superconducting states. The prediction of the electron-hole asymmetry in doping is quite interesting and will give insight into the existing experimental results.

We would like to thank the reviewer for summarizing our work nicely. In fact, our work was motivated by the experimental indication of the direct role of charge fluctuation.

The manuscript is intelligibly written. The model includes many parameters such as several transfer integrals and Coulomb interactions, U and V ; however, the authors neatly fix the parameters and relate U and V to pick up the essential features of the model. I think that this manuscript deserves publication in Nature Communications.

We thank the reviewer for the recommendation for publication in Nature Communications. As the reviewer pointed out, the model includes many parameters and we fix some of them to be appropriate for κ -(ET)₂X. This enables us to give a general discussion which is not limited to a special case.

A minor comment--- can authors comment on the magnetic ground state of 3-fold CO-2?

We have tried several magnetic orderings that coexist with 3-fold CO-2 but none of them are stabilized. We think that the reasons are as follows: First, the CO pattern in the hole-rich sites forms a triangular-like structure and therefore the spin degree of freedom is fully frustrated. Furthermore, the distance between hole-rich sites are much longer than the original intermolecular bonds and therefore the effective magnetic exchange couplings are quite small. We have added the explanation in the main text and in the caption of Figure 2 [list of changes (4)].

Reviewer #2 (Remarks to the Author):

This manuscript by Watanabe et al. employed a microscopic model to investigate the pairing symmetry of unconventional superconductivity in molecular conductors κ -(BEDT-TTF)₂X in both electron-doped and hole-doped sides. It is currently an important issue due to the experimental realization of superconductivity in this family of strongly correlated systems. The microscopic model employed by the authors is the extended Hubbard model with both onsite and intersite Coulomb interactions. By doing variationally Monte Carlo calculations, they found apparent asymmetry between the electron-doped and hole-doped sides of the phase diagram. The pairing symmetry is d_{xy} in the hole-doped side while it is extended $s+d_{x^2-y^2}$ in the electron-doped side. They also provided some explanations about the asymmetry. To me, this work is important and timely due to the experimental relevance although the method and model employed are not novel. I intend to recommend its publication in Nat. Comm. after the authors could address the comments/questions I have below.

We acknowledge the reviewer's comments on the novel competition between different pairing symmetry appearing asymmetric with respect to the doping rate, and for appreciating the importance of our work. Although the method and the model are both utilized in the past, as pointed out by the reviewer, their combination here under finite doping is applied for the first time and brings about novel results distinct from previous studies.

1) In the hole-doped side, superconductivity is not the most favored state in the model studied by the authors. I wonder if the SC with d_{xy} pairing symmetry can be realized as a ground state by using another set of parameters in the extended Hubbard model.

We acknowledge the reviewer for the important suggestion. We have also calculated the cases for $V/U=0.4$ and 0.55 (the present study is for $V/U=0.5$), and also for varied dimerization (artificial

change of t_{b1}). However, the d_{xy} -wave SC is always masked with the DAF phase and is not the ground state, at least in the parameter range we tried. This is owing to the fact that the AF spin fluctuation in the paramagnetic metallic state is not strong enough, in contrast with the square lattice models, e.g. for the cuprates, because of the peculiar triangular-like structure of κ -(ET)₂X. The hole doping partially releases the geometrical frustration and the AF spin fluctuation is certainly enhanced but is not enough to stabilize SC. However, it is possible that the “hidden” SC phase appears if the DAF phase is destabilized by some reason (e.g., disorder effect associated with doping, or phase separation). Therefore, we think that it is worthwhile to study the most favored SC phase even if it is a metastable state. We added sentences about this point [list of changes (5)].

2) In the electron-doped side, the SC pairing symmetry is extended $s+d_{x^2-y^2}$ which breaks both charge-U(1) symmetry and lattice rotational symmetry. I wonder if the authors could comment on its finite temperature behaviors, namely whether there are two transitions or one transition as the temperature is lowered.

The extended- $s+d_{x^2-y^2}$ -wave SC we discuss here does not break the lattice rotational symmetry. Although the notations might be confusing (we followed the conventional use of them), both s and $d_{x^2-y^2}$ belong to the same A_{1g} irreducible representation of the D_{2h} point group of this material. Therefore, these components can “coexist” to accommodate the orthorhombicity of the system. In other words, it should be described by a single order parameter. Then, we expect that only one transition occurs at finite temperature, consistent with experimental observations.

We also note that, depending on the coordinate system, the notation of xy and x^2-y^2 are sometimes exchanged in the literature. For example, Powell and McKenzie (PRL 98, 027005, 2007) denote the same symmetry as “(extended-) $s+d_{xy}$ ” since they choose the coordinate system following the convention in the dimer model.

3) To what extent the asymmetry between hole-doping and electron-doping depends on the model parameters. Namely, is this asymmetry robust enough?

Yes, the electron-hole asymmetry is robust. As shown in Fig. 1b, a van Hove singularity is located in the hole-doped side. This, together with the electron correlation effect, is the main reason for the electron-hole asymmetry. Although the exact location of the van Hove singularity depends on the model parameters, it always locates in the hole-doped side as long as the realistic conditions are considered. Therefore, this asymmetry itself is robust against the realistic parameter changes

and is relevant for κ -(ET)₂X in general. We added sentences about this issue [list of changes (7)].

Reviewer #3 (Remarks to the Author):

summary:

Based on Variational Monte Carlo (VMC) calculations the authors present a theoretical analysis of the phase diagram of the extended Hubbard model on a two-dimensional lattice representing the lattice structure of κ -(ET)₂X molecular conductors. The authors investigate the appearance of various ground-state phases as a function of onsite-Coulomb-repulsion strength and electron filling. Metallic, superconducting and insulating phases of various kinds (antiferromagnetic and different type of charge ordered phases) are presented and discussed. The authors point to the asymmetry in the phase diagram between hole doping and electron doping the lattice and its relation to recent experimental observations in κ -(ET)₂X systems where carrier doping was achieved either chemically (Ref. 40-43 in the manuscript) or by field-effect doping (Ref. 44, 45 in the manuscript).

We appreciate the reviewer's careful reading of the manuscript. As pointed out by the reviewer, this is the first theoretical work to examine the doping effect in κ -(ET)₂X. We have shown that the carrier doping provides much richer physics than the bandwidth-control case. The mechanism of emergent phases including SC is clarified in terms of the change of the degree of frustration. The importance and novelty of the present work will be explicitly displayed in the following responses.

originality and discussion:

the family of ET charge transfer salts in a κ arrangement has been experimentally and theoretically intensively studied as a representative of Mott-Hubbard physics with a rich phase diagram including Mott insulator, bad metal and superconductor phases as a function of chemical and/or external pressure.

More recently the possibility of carrier doping, as mentioned above, has evidenced a significant electron-hole doping asymmetry. The extended Hubbard model, as studied by the authors, has been proposed in the past by many authors, including some of the authors of the manuscript, to be an adequate model to study these materials and there exist already many investigations on the ground-state phases of this model for various interaction strengths (onsite and nearest-neighbor)

performed by some of the authors as well as by other groups by means of VMC and a few other many-body methods. The novelty of the present manuscript is that the authors extend the study to carrier filling away from the 3/4 filling that covers regions in phase space now accessible to experiment and they observe an electron-hole asymmetry.

We would like to thank the reviewer for appreciating the novelty of our results. First, we would like to note that many of the previous works on κ -(ET)₂X have been done using the 1/2-filled Hubbard model (sometimes called as the dimer model) that neglects the charge degree of freedom within the dimers. They are very successful in describing the Mott-Hubbard physics, as the reviewer pointed out. However, concerning the description of superconducting states, including its stability in competition with other symmetry-broken phases, there is a serious controversy. Recent numerical studies including VMC even deny its stability in this model. Therefore, the proper choice of the model incorporating the role of charge fluctuations, in addition to the spin fluctuations, is essential to understand the physics of κ -(ET)₂X, as we have done here in this study. This is one of the novelties of our study. [see list of changes (6)]

Whether these results are enough to publish the manuscript as a nature communications is not completely clear. I view the present manuscript as an extension of previous results that would be appropriate in a more specialized journal.

First, as we described above, this work is not along the direction of conventional Mott-Hubbard physics that can be captured in the 1/2-filled Hubbard model. Second, as the reviewer pointed out, now there are a lot of studies for the extended Hubbard model that focus on the interplay between the spin and charge degrees of freedom, and geometrical frustration at 3/4 filling. However, no study has been reported for the filling-control case before our study here. Owing to the non-perturbative variational Monte Carlo calculations, our study reveals novel results, not obtained by a simple extension of previous studies. Let us elaborate this in the following.

First, by analyzing the spin/charge structure factors, we found that the degree of geometrical frustration can be controlled by doping. This has a significant consequence to the electro-hole doping asymmetry of the phase diagram, particularly, the doping asymmetry of the symmetry of SC. This is difficult to realize by simply controlling the bandwidth (applying pressure or substitution of anions) and gives conceptually new perspective not only to κ -(ET)₂X but also to frustrated systems in general.

Next, a direct comparison with cuprate superconductors becomes possible only through this study.

It has been a long-standing problem whether physics in these two classes of systems is the same. Our numerical study clearly shows that the electron-hole asymmetry in $\kappa\text{-(ET)}_2\text{X}$ is fundamentally different from that of cuprates, especially, for the SC phase. The difference results from the charge degree of freedom within the dimers and the geometrical frustration in $\kappa\text{-(ET)}_2\text{X}$, which are absent in cuprates. The difference has become evident only when the mobile carriers are introduced into a Mott insulator at $3/4$ filling.

We have added the sentences to emphasize these two points in the revised version of the manuscript [list of changes (1)].

The manuscript is well written, however there are a few issues to consider:

- while the authors cite previous works related to the model considered and results obtained, there is no discussion on the comparison of phases obtained in the present work with respect to what other groups have done. In other words, the cited work is not commented or compared although the authors are using similar methods to some of them and explore similar phase regions. How do their results compare with previous ones, where are the similarities and where are the differences in the common regions studied?

We acknowledge the reviewer for pointing this out. We admit the reviewer that the discussion on the comparison between our work and the previous works was lacking in the previous version of the manuscript, which blurred the novelty of our results.

As for the SC, the competition between the extended- $s+d_{x^2-y^2}$ -wave and the d_{xy} -wave in the undoped case has been studied for decades. We added the discussion on this issue in the revised version of the manuscript [list of changes (6) and (8)]. When the intersite Coulomb interactions are absent, these two SC are in severe competition and the expected SC transition temperatures are too low. When the intersite Coulomb interactions are introduced, both of them are enhanced in their stability, but the extended- $s+d_{x^2-y^2}$ -wave is slightly more favored. Although they are still competing with each other, the carrier doping releases this competition due to the change of the degree of geometrical frustration. This is our main result and distinct from the previous studies.

Other phases such as DAF and PCO have also been proposed in several previous studies for the undoped case. The DAF phase is the conventional Neel ordered Mott insulating phase, which is common in the $1/2$ -filled Hubbard model and the $3/4$ -filled extended Hubbard model. The PCO state is the charge-ordered insulating state in the dimerized lattice structure, which is characteristic

of the 3/4-filled extended Hubbard model. These phases appear in our study with the same symmetry, but the non-trivial point that we found in this study is that they survive under finite doping level as metallic phases. The stability of DAF depends on the AF spin fluctuation, which is enhanced by the hole doping and is suppressed by the electron doping. On the other hand, we found that the PCO phase is rapidly suppressed when the electron density deviates from the commensurate filling at $n_{\text{hole}}=1/2$ (i.e., undoped case at 3/4 filling).

The 3-fold CO states have also been discussed in different contexts such as frustrated triangular lattice extended Hubbard models, but not in the context of $\kappa\text{-(ET)}_2\text{X}$ except that our previous study has found that one of the 3-fold CO states appear in the large V_{ij} region at 3/4 filling.

We added sentences to describe these points in the revised version of the manuscript [list of changes (2), (3), and (8)].

- how would temperature affect the results?

The variational Monte Carlo method used here in this study is only for a ground state at zero temperature, but not for finite temperatures. However, by referring to results such as DMFT, we can naively infer that the phases extend up to their characteristic energy scales roughly set by the variational energy differences among different phases (while reduced by the low dimensionality and quantum fluctuation). Therefore, our main results will not change qualitatively at low temperatures below such energy scales.

- the authors mention that superconductivity is not the ground-state in the hole-doped side. Which is the ground state? and how is then the phase diagram modified?

The SC phase is not the ground state but the metastable state in the hole-doped side. Fig. 2f shows the most favored SC symmetry, ignoring other ordered phases. The ground state phase diagram is given in Fig. 2a and is not modified. The point showing Fig. 2f is that although the SC is not the ground state, it has finite condensation energy (namely, it has lower energy than normal metal) in a wide region of the phase diagram and competes with other ordered phases. It is possible that the “hidden” SC phase appears if the DAF phase is destabilized by some reason (e.g., disorder effect associated with doping, or phase separation). Therefore, we think that it is worthwhile to study the most favored SC phase even if the SC is a metastable state. In the revised version of the manuscript, we added sentences to explain this point better [list of changes (5)].

In summary, from the reasons above, we strongly believe that our study is not the simple extension of the previous results. Some of the results obtained here are general and can be applied beyond the molecular systems κ -(ET)₂X. Therefore, we believe that our results attract general interests and the manuscript deserves to be published in Nature Communications after the appropriate revision that we have made.

Sincerely yours,

Hiroshi Watanabe, Hitoshi Seo, and Seiji Yunoki

REVIEWERS' COMMENTS:

Reviewer #1 (Remarks to the Author):

The authors' reply to my question is satisfactory. They added their reply to the main body of text. I think that this additional statement/discussion is quite intriguing and important for further studies on the magnetism of the system in question.

Kazushi Kanoda

Reviewer #2 (Remarks to the Author):

The authors have addressed the questions raised by all three reviewers reasonably well and have revised the manuscript accordingly. I have no further questions/comments. As I mentioned in my previous report, the present work is quite important and is also timely because of its experimental relevance. I now recommend its publication in Nature Communications.

Reviewer #3 (Remarks to the Author):

The authors made a thorough revision of the manuscript and provided convincing arguments to my comments and therefore I recommend publication of the manuscript in Nature Communications.

Response to Reviewer Comments

Reviewer #1 (Remarks to the Author):

The authors' reply to my question is satisfactory. They added their reply to the main body of text. I think that this additional statement/discussion is quite intriguing and important for further studies on the magnetism of the system in question.

Kazushi Kanoda

Reviewer #2 (Remarks to the Author):

The authors have addressed the questions raised by all three reviewers reasonably well and have revised the manuscript accordingly. I have no further questions/comments. As I mentioned in my previous report, the present work is quite important and is also timely because of its experimental relevance. I now recommend its publication in Nature Communications.

Reviewer #3 (Remarks to the Author):

The authors made a thorough revision of the manuscript and provided convincing arguments to my comments and therefore I recommend publication of the manuscript in Nature Communications.

We would like to thank all reviewers for their valuable and insightful comments and suggestions to our manuscript. We appreciate the positive comments for publication from them.

Sincerely yours,

Hiroshi Watanabe, Hitoshi Seo, and Seiji Yunoki